# Knowledge and Attitudes toward Autism Spectrum Disorder in Saudi Arabia

**DOI:** 10.3390/ijerph19063648

**Published:** 2022-03-19

**Authors:** Hamad S. Alyami, Abdallah Y. Naser, Mohammad H. Alyami, Salem H. Alharethi, Abdullah M. Alyami

**Affiliations:** 1Department of Pharmaceutics, College of Pharmacy, Najran University, Najran 66262, Saudi Arabia; mhalmansour@nu.edu.sa (M.H.A.); abooad2211@gmail.com (A.M.A.); 2Department of Applied Pharmaceutical Sciences and Clinical Pharmacy, Faculty of Pharmacy, Isra University, Amman 11622, Jordan; abdallah.naser@iu.edu.jo; 3Department of Biological Science, College of Arts and Science, Najran University, Najran 66262, Saudi Arabia; shalharthi@nu.edu.sa

**Keywords:** attitudes, autism, knowledge, Saudi Arabia

## Abstract

Aims: The diagnosis of autism spectrum disorder (ASD) is not easy as there is no direct test that exists to establish such a diagnosis. Increasing community and healthcare professional knowledge of the ASD spectrum is crucial because it will encourage parents of ASD children to seek screening and diagnosis from a specialist, allowing for better early detection and treatment. This study explored the knowledge of the general population in Saudi Arabia regarding ASD and assessed variables associated with an accurate understanding of ASD. Methods: A total of 769 participants were involved in this cross-sectional study, which was conducted in Saudi Arabia between November 2021 and February 2022 using an online survey tool to explore the knowledge of the general population in Saudi Arabia regarding typical child development and ASD. A binary logistic regression analysis was used to determine factors affecting participants’ knowledge of autism. Results: Overall, the study participants showed a weak level of knowledge about autism with a mean score of 5.9 (SD: 3.1), comprising 34.7% of the total maximum obtainable score. Participants with a middle income category of 5000–7500 SR are less likely to be knowledgeable about autism compared to others (OR: 0.60 (95% CI: 0.39–0.92)) (*p*-value = 0.020). Conclusion: The participants in our study showed limited knowledge about autism. Government funds should be made available to facilitate educational services for ASD children. More funding and resources should be allocated by the government to provide assistance for children with special needs, and changes in public facilities are required to meet the demands of ASD patients. Through an informed educational effort, various media platforms should assist in improving the community’s understanding of ASD and their attitude toward ASD patients. Educational campaigns should focus on enhancing the public’s knowledge about ASD treatment and etiology.

## 1. Introduction

Autism spectrum disorder (ASD) is a childhood developmental disorder. It is characterized by multiple social, behavioral, and communication difficulties. ASD is accompanied by anatomical and functional changes in the brain of an ASD patient [1]. ASD could have varying degrees of severity, with some cases being moderate and others being severe. Severe cases necessitate a great deal of assistance and ongoing support. Patients with ASD have a tendency to repeat certain behaviours and might not want change in their daily routine [1].

This disorder affects one out of every 44 children, and it can affect people of all races and ethnic groups. As a result, it is one of the most frequent developmental impairments among children [2]. There has been an increase in the incidence of ASD in the last two decades, and one possible reason for this increase is that, while this condition is a lifelong disorder, there is greater awareness of it today than in the previous two decades, and as a result, caregivers of ASD children are beginning to seek screening and diagnosis from a specialist [3].

ASD diagnosis is problematic because no direct tests, such as a blood test or particular biomarkers in the body, are available to confirm the diagnosis. A diagnosis for ASD is based on observing the child’s behaviour [3]. In 2013, the Diagnostic and Statistical Manual of Mental Disorders (DSM)-5 of the American Psychiatric Association published their criteria for the diagnosis of ASD spectrum, adding pervasive developmental disorder and Asperger’s disorder to the ASD spectrum. Before that, they were classified as different subgroupings of the ASD spectrum, but now they consider autistic syndrome, Asperger’s disorder, and pervasive developmental disorder to be part of the ASD spectrum. According to the Diagnostic and Statistical Manual of Mental Disorders (DSM)-5, ASD is linked to a variety of mental disorders, including attention deficit hyperactivity disorder (ADHD), social anxiety disorder, depression, and intellectual disability [4,5,6].

The age at which a child is diagnosed varies from case to case; some children are diagnosed as early as 18 months, but the majority of them require a further 18 months to confirm the diagnosis, which unfortunately causes a delay in treatment commencement [7]. Early treatment enhances the verbal and cognitive abilities of ASD children [8]. When compared to mild symptoms, more acute symptoms are a cause of early diagnosis [7]. One factor contributing to the delay in confirming diagnoses in Saudi Arabia is a lack of information and awareness among both the community and healthcare staff [9].

To diagnose ASD patients, a variety of screening instruments are utilized, some of which are widely used, such as the Modified Checklist for Autism in Toddlers (M-CHAT), and others that are less often used, such as the Communication and Symbolic Behavior Scales (CSBS) [10]. In diagnosing ASD, the Autism Diagnostic Observation Schedule-2 (ADOS-2) is considered the “gold standard”. It is a semi-structured, standardized examination of social interaction, play, and imaginative material used for people suspected of having ASDs, ranging in age from 12 months to adulthood [11].

Some of the risk factors for developing ASD include genetic and environmental variables [12]. Some medicines taken by the mother during pregnancy, such as valproic acid, have been linked to the development of ASD in children [13]. ASD is more likely to affect children born to older parents. Furthermore, the age of the pregnant lady, i.e., if the mother is over 40 years old, has an impact on the development of ASD [14]. Additionally, whether the mother was pregnant while suffering from hypertension or a viral or bacterial infection has a bearing on the development of the disorder [15]. A child’s brother or sister is more likely than other siblings to have an ASD if he or she has an ASD [16,17,18]. A genetic and chromosomal disorder, such as the Fragile X chromosome, can potentially cause ASD [4]. Besides, changes in the GABAergic, glutamatergic, serotonergic, and dopaminergic systems have been linked to ASD after exposure to neurotoxic chemicals [19].

There are many myths and misconceptions about ASD. For example, some people believe that vaccines are responsible for the development of ASD [16]. According to the World Health Organization website (WHO), there is no documented link between a vaccine and the development of ASD, and there is also no link between the MMR vaccine and the development of ASD disease [20].

There is no medical treatment for ASD that is able to cure the disorder. All the child’s caregivers can do is to begin using special behavioral therapies as soon as the diagnosis is confirmed to help ASD patients develop new abilities such as talking, looking in their eyes, playing with peers, walking, and socializing [21]. The cost of ASD management is considered high because there are direct and indirect costs to the family, such as health costs and the coast of extra educational support that is required, particularly for children with intellectual disabilities, which will place a financial strain on the family and, in many cases, result in mothers losing their jobs [22,23].

As previously noted, increasing community and healthcare worker understanding of the ASD spectrum is critical in order to enhance both early diagnosis and treatment, which will lead to a better prognosis. As a result, determining the general public’s awareness of typical infant development and ASD in Saudi Arabia has become a critical issue to investigate in order to pinpoint knowledge gaps and provide appropriate intervention.

## 2. Method

### 2.1. Study Design

A cross-sectional study was conducted in Saudi Arabia between November 2021 and February 2022 using an online survey tool to explore the knowledge of the general population in Saudi Arabia regarding typical child development and ASD.

### 2.2. Questionnaire Tool

A previously validated questionnaire was adapted and utilized to achieve the study objectives, developed by Liu et al. [24]. The questionnaire tool is comprised of 43-items. The first section is comprised of seven items and describes the demographic characteristics of the study participants. The second section is comprised of 17 items in yes/no format and explores participants’ knowledge of ASD. Each correct answer was given a score of one, and the total achievable knowledge score is seventeen. The higher the score, the more knowledgeable the participant is. Participants’ knowledge of the symptoms and behavior of children with ASD, typical child development, the importance of early diagnosis in treatment success, the causes and risk factors of ASD, the proper treatment technique, the etiology of this disorder, and the incidence of this disorder were all tested.

The third section is comprised of 10 items on a 5-point Likert scale (ranging between 1 “strongly disagree” and 5 “strongly agree”) that explored participants’ attitudes towards care and advocacy for children with ASD. Participants’ attitudes towards autistic children’s special needs, educational and teaching requirements, government funding, resource distribution for autistic children, and insurance policy coverage were investigated.

The fourth section was comprised of four items on a 5-point Likert scale (ranging between 1 “strongly disagree” and 5 “strongly agree”) that explored participants’ interest and perceived efficacy toward ASD. In the last section, participants were asked about their awareness of interventions and approaches devoted to the care of individuals with ASD using 5-items of yes/no format.

### 2.3. Ethical Considerations

The study protocol was reviewed and ethical approval was granted by the Deanship of the Scientific Research Ethics Committee at Najran University, Najran, Saudi Arabia (443-41-33598-DS).

### 2.4. Statistical Analysis

The data for this study was analyzed using Statistical Packages for Social Sciences version 27 (SPSS Inc., Chicago, IL, USA). The frequency and percentage of categorical data were reported. The data were verified for normality using histogram and normality measures, which revealed that they were normally distributed. The mean (SD) was used to present continuous variables such as the participants’ autism knowledge score. The mean autism knowledge score was compared between different demographic groups using an independent sample *t*-test and a one-way analysis of variance (ANOVA). Fisher’s least significant difference (LSD) post hoc test was used to identify the source of significant variation within each group. The mean participants’ autism knowledge score (5.9) was used as a cut-off point in the binary logistic regression analysis to determine factors affecting participants’ knowledge of autism. A confidence interval of 95% (*p* < 0.05) was applied to represent the statistical significance of the results, and the level of significance was assigned as 5%.

## 3. Results

### 3.1. Participants’ Demographic Characteristics

A total of 769 participants were involved in this study. The majority of them (86.1%) were males. Most of the participants (68.3%) were aged under 34 years. Around half of the participants (49.3%) were single. More than half of them (61.9%) reported having a bachelor’s degree. A total of 43.6% of the participants reported working outside the healthcare sector. Around 44.0% of them reported that their monthly income was 7500 SR or above. When the participants were asked whether they had first-degree relatives with autism, 16.0% of them confirmed that. Table 1 below describes the demographic characteristics of the study participants.

### 3.2. Knowledge about Autism

Overall, the study participants showed a weak level of knowledge about autism with a mean score of 5.9 (SD: 3.1), comprising 34.7% of the total maximum obtainable score. The mean autism knowledge score showed a statistically significant difference between participants based on their employment status and having a first-degree relative with autism (*p* ≤ 0.05). Table 2 shows the mean autism knowledge score stratified by demographic characteristics.

Participants’ responses to knowledge items are presented below in Figure 1.

In Figure 1, the percentage of participants who answered the knowledge items correctly ranged between 11.7% and 73.0%. The lowest percentages of correct answers were items related to knowledge about ASD treatment and etiology. Only 11.7% of the participants understood that ASD is not curable, even if it was diagnosed early and the appropriate intervention was provided. Only 14.0% of the participants knew that ASD treatment does not cure autism. The highest percentages of correct answers were items related to knowledge about the symptoms and behavior of ASD children. The majority of the participants (73.0%) were aware that ASD children require special education services at school; 54.7% of them knew that children with ASD often present with speech and language delays between 2 and 3 years of age, and 53.3% knew that children with ASD often do better with visual input than with auditory input.

Binary logistic regression analysis identified that participants with an income category of 5000–7500 SR are less likely to be knowledgeable about autism compared to others (OR: 0.60 (95% CI: 0.39–0.92)) (*p*-value = 0.020), Table 3.

### 3.3. Attitudes towards the Care and Advocacy for, Children with Autism

Ten items were utilised to explore the attitudes of participants towards the care and education of, and advocacy for, children with autism. The most commonly agreed upon items were that government funding should be made available to facilitate staff employment in preschools to meet the needs of these children (79.1%), the government should allocate more resources for the provision of services for children with special needs (76.6%), insurance policies should be amended to include coverage for developmental disorders as chronic disabilities (74.3%), and all preschools should have special education teachers and therapists to provide services to children with special needs attending class there (72.7%). Figure 2 below shows participants’ attitude towards the care and education of, and advocacy for, children with autism.

### 3.4. Interest and Perceived Efficacy

Four items were utilised to explore participants’ interest and perceived efficacy towards autism management. The participants showed the highest degree of agreement that there is a need for change in public facilities to meet the needs of autistic patients (72.2%), followed by the idea that they are keen to be a partner in the treatment and improvement of autistic patient’s conditions, for example, the use of visual aids and medicines (57.1%). Figure 3 presents the degree of agreement concerning participants’ interests and perceived efficacy towards autism management.

### 3.5. Awareness of Approaches Devoted to the Care of Individuals with Autism

When the participants were asked about their awareness of the approaches devoted to the care of individuals with autism, around half of them (51.4%) confirmed that they were aware of relationship development programs, followed by auditory integration therapy (47.9%). Table 4 below shows participants’ awareness of approaches devoted to the care of individuals with autism.

## 4. Discussion

This study aimed to explore the knowledge of the general population in Saudi Arabia regarding ASD and assess variables associated with an accurate understanding of ASD. Moreover, this study explored participants’ attitudes towards autism and their awareness of approaches devoted to the care of individuals with autism. The key findings of our study are: (1) our study participants showed a weak level of knowledge about autism with a mean score of 5.9 (SD: 3.1) (34.7%); (2) participants with a middle income category of 5000–7500 SR are less likely to be knowledgeable about autism compared to others; (3) concerning participants’ attitude towards the care and advocacy of children with autism, the most commonly agreed upon items were that government funding should be made available to facilitate staff employment in preschools to meet the needs of these children, the government should allocate more resources for the provision of services for children with special needs, insurance policies should be amended to include coverage for developmental disorders as chronic disabilities, and all preschools should have special education teachers and therapists to provide services to children with special needs attending class there; (4) participants showed the highest degree of agreement that there is a need for changes in public facilities to meet the needs of autistic patients, followed by that they are keen to be a partner in the treatment and improvement of autistic patient’s conditions, for example, the use of visual aids and medicines, and (5) participants were more aware of relationship development programs and auditory integration therapy as approaches devoted to the care of individuals with autism.

A previous study conducted in the United States found that around 90% of participants had appropriate knowledge of ASD in the domains of diagnosis and symptoms, etiology, and therapy [25]. However, only 34.7% of participants had good knowledge of ASD, which is similar to the results of a prior study conducted in Saudi Arabia [26]. Despite having heard about ASD before, the participants in this study had a poor understanding of the disorder and many misconceptions about it [26]. Previous research in the Middle East region, including Lebanon and Oman, demonstrated that students and professionals have insufficient knowledge about ASD [27,28]. In a previous Australian survey, the majority of participants had heard of ASD disorder, but there was a knowledge gap about the condition’s etiology and impact [29]. The majority of Australians were aware of ASD, but they mistook it for a reaction to the MMR vaccine [29]. The World Health Organization (WHO), on the other hand, said that no evidence linking any vaccine to autism exists [30]. Another study found that participants misunderstood the treatment, believing that children with ASD are unable to attend public school [31]. As previously said, a variety of factors contribute to the development of this condition, some of which are genetic in origin and others which are environmental in nature [15,32]. In another African study, not just among communities but also among health care providers and educators, there was a lack of understanding [31].

Males made up about 86% of the participants in this study. Previous research has found that females are more aware of autism disorder than males, according to studies that compared knowledge by gender. One possible reason for this is that females are more interested in studying diseases than males [3,15,26,29,33]. One of these studies indicated a substantial difference in knowledge between males and females, as well as a stronger knowledge of specific psychiatric illnesses in females [33]. Furthermore, the bulk of the participants were under the age of 34 years. This finding reveals that individuals in the younger age groups are unaware of ASD. Similarly, in a prior study, older participants had more experience than younger participants [26]. According to one study, while there is a difference between male and female knowledge, neither their age nor their educational degree has an impact on the impact of their knowledge [33]. A previous study that examined the knowledge about ASD in China reported that gender and socioeconomic status were important demographic variables that had an influence on ASD knowledge [25]. Many variables contribute to the difference in the level of public knowledge about ASD. These include cultural differences and information sources (whether internet, social media or TV) [25].

A previous study on the understanding of ASD among healthcare personnel in Saudi Arabia found that half of them had a moderate to poor level of knowledge about ASD, with the biggest gaps in their knowledge regarding the disorder’s origins and comorbidities [9]. Healthcare providers and educators, according to another study [34], needed to learn more about ASD. The highest level of expertise is held by pediatricians and psychiatrists [9,35].

In our study, we found that the highest percentages of correct answers were items related to knowledge about the symptoms and behavior of ASD children. The lowest percentages of correct answers were items related to knowledge about ASD treatment and etiology. Confirming the findings of a previous study that examined the knowledge about ASD in China and the USA, pharmaceutical treatment for ASD is often misunderstood by the general public [25]. Knowing the risk factors for ASD is crucial in lowering the chances of having children with the disorder [36]. Early detection of the disorder can lead to better outcomes [29], as it can also lead to significant changes in a child’s linguistic, cognitive, and adaptive behavior [37,38]. Furthermore, early intervention will improve the child’s social behavior and daily skills before the child reaches the age of four [39], keeping in mind that early detection is not always a simple process that confirms the diagnosis, as it is highly dependent on the level of education of the HCP and the execution of ASD guidelines [40]. According to a previous study, it could take up to two years between the initial observation of the child’s behavior and the confirmed diagnosis [40].

Our study found that there was a significant difference in knowledge between people who had a first-degree relative with autism and those without. This finding is consistent with earlier research [3]. Previous research indicated that participants who had a direct interaction with ASD patients had better knowledge because they had more experience [3].

The majority of participants agreed that government funding should be made available to help preschools hire more staff to meet the needs of these children, that the government should allocate more resources to providing services for children with special needs, and that all preschools should have special education teachers and therapists to provide services to children. Almost all prior studies concluded that the government had a vital role to play in increasing awareness of ASD and assisting families with autistic patients [9,29,31,41,42,43,44].

One recommendation was that the government launch a nationwide awareness campaign to raise awareness [9]. According to a 2015 Australian study, the government provided the necessary support only for younger children with ASD under the age of 5 years, but not for older adolescents between the ages of 13 and 17, resulting in a decline in their secondary school education and future work opportunities [29]. This had a significant impact on autistic students’ employment, as researchers revealed that autistic patients were employed at a rate of three times less than people with other disabilities and six times less than people without disabilities [29]. Another study found that a lack of assistance from school professionals in an education system caused ASD students to become socially isolated and leave their schools [42]. Government funding was helpful in educating parents and teachers of children with ASD [41,44,45].

Because the cost of treatment for developmental disorders is believed to be significant, the majority of the participants felt that insurance policies should include coverage for developmental disorders. According to a previous study, the cost of direct and indirect treatment of ASD patients over their lifetime in the United Kingdom and the United States of America was $2.4 million and $2.2 million, respectively, if the ASD patient had an intellectual disability, compared to $1.4 million and $1.4 million, respectively, if the ASD patient did not have an intellectual disability [22]. According to another study, the cost of special education services for ASD children was 76% greater than for non-ASD students, which was just 7% higher, and ASD students required roughly $8610 more in school costs than non-ASD students [23]. Early autism illness diagnosis and intervention have been shown to have a significant impact on treatment outcomes, such as improving efficacy and reducing the severity of the disease, particularly in cognitive and behavioral aspects [37,38]. Increasing awareness of ASD indicators and symptoms will enhance early diagnosis and, as a result, treatment seeking. Although there is currently no treatment for ASD, studies have indicated that certain behaviors can assist pregnant women in preventing autism. According to research published in the New England Journal of Medicine, autistic children’s brain development abnormalities begin as early as the second trimester. The following recommendations can help pregnant women prevent autism in their growing fetus beginning at conception, such as reducing toxin exposure, eating a healthy diet, and visiting their family physician and obstetrician on a regular basis [46].

According to the findings of our study, there is a knowledge gap in the community on ASD, particularly among males and young participants. Non-communicable disorders, such as psychiatric and developmental disabilities, should receive more attention from public health professionals since they have a significant influence on community productivity. This places pressure on both national policymakers and healthcare practitioners to improve overall knowledge and reduce misconceptions. Awareness campaigns, the distribution of leaflets and booklets, and the use of multimedia platforms such as Facebook, Twitter, and local television channels are all recommended options to enhance the knowledge of the community about ASD and improve their attitude towards ASD patients.

There are some limitations to this study. The cross-sectional study design prevents the establishment of causality between study variables. The majority of the study participants (86.1%) were males, which could have affected the generalizability of our findings.

## 5. Conclusions

The participants in our study showed limited knowledge about autism. Government funds should be made available to facilitate educational services for ASD children. More funding and resources should be allocated by the government to provide assistance for children with special needs, and changes in public facilities are required to meet the demands of ASD patients. Through an informed educational effort, various media platforms should assist in improving the community’s understanding of ASD and their attitude toward ASD patients. Educational campaigns should focus on enhancing the public’s knowledge about ASD treatment and etiology. 

## Figures and Tables

**Figure 1 ijerph-19-03648-f001:**
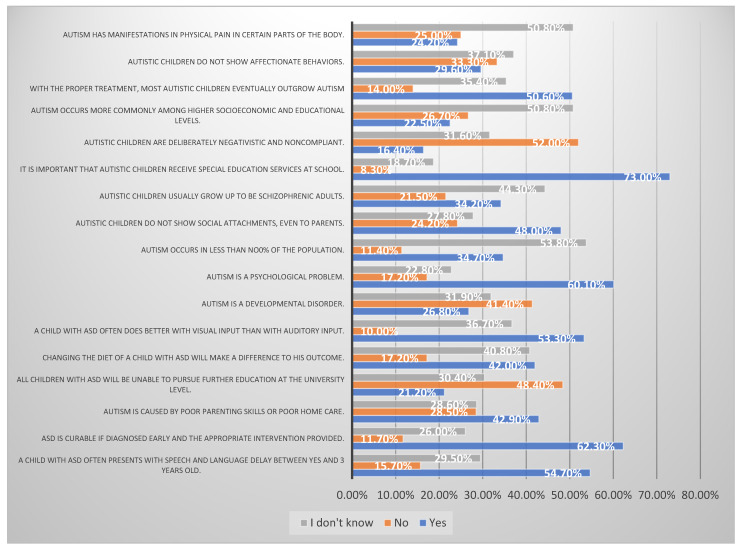
Participants’ responses for knowledge items.

**Figure 2 ijerph-19-03648-f002:**
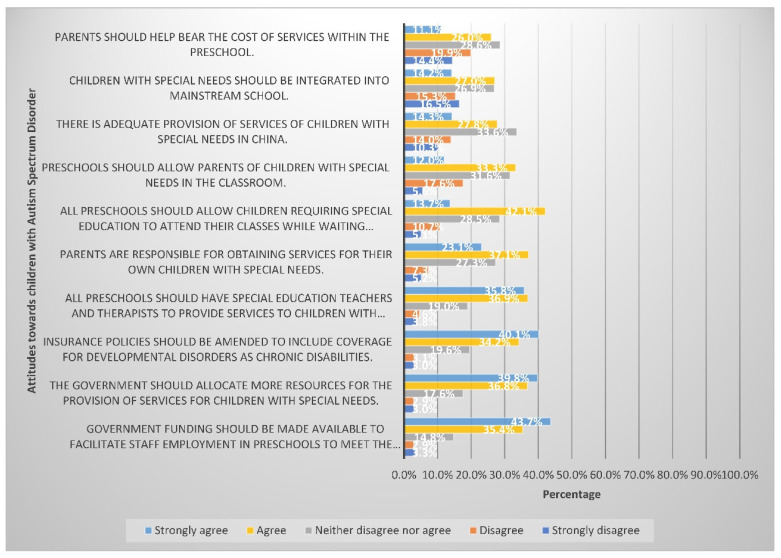
Attitudes towards the care and education of, and advocacy for, children with autism.

**Figure 3 ijerph-19-03648-f003:**
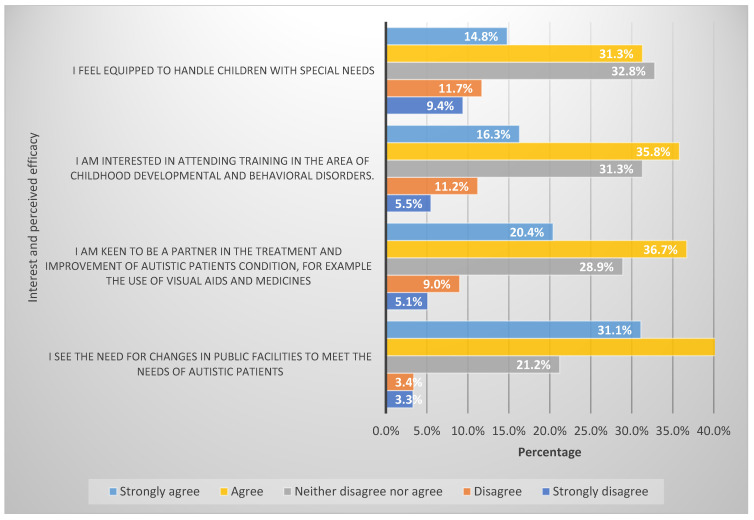
Participants’ interest and perceived efficacy towards autism management.

**Table 1 ijerph-19-03648-t001:** Participants demographic characteristics.

Demographic Variable	Frequency	Percentage
Gender		
Male	662	86.1%
Age category		
18–23 years	147	19.1%
24–29 years	238	30.9%
30–34 years	141	18.3%
35–39 years	85	11.1%
40–45 years	87	11.3%
46 years and above	71	9.2%
Marital status		
Single	379	49.3%
Married	345	44.9%
Divorced	34	4.4%
Widowed	11	1.4%
Education level		
Secondary school or lower	218	28.3%
Bachelor degree	476	61.9%
Higher education	75	9.8%
Employment status		
Working outside the healthcare sector	335	43.6%
Unemployed	153	19.9%
Student	134	17.4%
Working inside the healthcare sector	108	14.0%
Retired	39	5.1%
Monthly income		
2500 SR or lower	245	31.9%
2500–5000 SR	91	11.8%
5000–7500 SR	94	12.2%
7500 SR and above	339	44.1%
Do you have a first-degree relative (father, mother, brother, sister, son, daughter, husband) with autism?		
Yes	123	16.0%

**Table 2 ijerph-19-03648-t002:** Mean autism knowledge score stratified by demographic characteristics.

Demographic Variable	Mean (SD)	*p*-Value
Gender		
Male	5.8 (3.1)	0.615
Female	6.0 (3.2)	
Age category		
18–23 years	5.9 (2.9)	0.933
24–29 years	5.8 (3.0)	
30–34 years	5.8 (3.1)	
35–39 years	6.1 (3.3)	
40–45 years	5.7 (3.4)	
46 years and above	6.0 (3.8)	
Marital status		
Single	6.0 (3.0)	0.758
Married	5.8 (3.4)	
Divorced	5.8 (2.9)	
Widowed	5.3 (2.6)	
Education level		
Secondary school or lower	5.7 (3.1)	0.198
Bachelor degree	5.8 (3.2)	
Higher education	6.5 (3.0)	
Employment status		
Unemployed	5.5 (3.2)	0.017 *
Student	5.9 (2.8)	
Retired	6.5 (3.6)	
Working inside the healthcare sector	6.7 (3.3)	
Working outside the healthcare sector	5.7 (3.1)	
Monthly income		
2500 SR or lower	5.9 (3.0)	0.104
2500–5000 SR	5.8 (3.2)	
5000–7500 SR	5.1 (2.9)	
7500 SR and above	6.0 (3.3)	
Do you have a first-degree relative (father, mother, brother, sister, son, daughter, husband) with autism?		
No	5.8 (3.2)	0.032 *
Yes	6.4 (2.7)	

* *p* ≤ 0.05.

**Table 3 ijerph-19-03648-t003:** Binary logistic regression analysis.

Demographic Variable	Odds Ratio of Being More Knowledgeable	95% CI
Gender		
Male (Reference group)	1.00	
Female	1.26	0.83–1.92
Age category		
18–23 years (Reference group)	1.00	
24–29 years	1.06	0.78–1.44
30–34 years	1.18	0.81–1.70
35–39 years	1.14	0.72–1.79
40–45 years	0.91	0.58–1.43
46 years and above	1.00	0.61–1.64
Marital status		
Single (Reference group)	1.00	
Married	1.10	0.83–1.47
Divorced	0.64	0.32–1.27
Widowed	0.99	0.30–3.26
Education level		
Secondary school or lower (Reference group)	1.00	
Bachelor degree	0.95	0.71–1.28
Higher education	1.43	0.87–2.33
Employment status		
Retired (Reference group)	1.00	
Working outside the healthcare sector	0.94	0.66–1.34
Unemployed	0.94	0.71–1.25
Student	0.85	0.58–1.23
Working inside the healthcare sector	1.47	0.97–2.24
Monthly income		
2500 SR or lower (Reference group)	1.00	
2500–5000 SR	1.06	0.68–1.64
5000–7500 SR	0.60	0.39–0.92 *
7500 SR and above	1.24	0.93–1.65
Do you have a first-degree relative (father, mother, brother, sister, son, daughter, husband) with autism?		
No (Reference group)	1.00	
Yes	1.35	0.91–2.00

* *p* ≤ 0.05.

**Table 4 ijerph-19-03648-t004:** Awareness of approaches devoted to the care of individuals with autism.

Are You Aware of the Following Intervention and Approaches Devoted to the Care of Individuals with Autism Spectrum Disorder?	
Relationship Development Intervention	51.4%
Auditory integration therapy	47.9%
Structured training	47.2%
Sensory integration therapy	45.8%
Applied behaviour analysis	36.7%

## Data Availability

Not applicable.

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
