# Peer review of "Knowledge and Attitudes toward Autism Spectrum Disorder in Saudi Arabia"

_ijerph, 2022, doi:10.3390/ijerph19063648_

Round 1

Reviewer 1 Report

The summary introduces from the beginning the acronym ASD without explaining its meaning. In the first sentence "diagnosis" is repeated.

Knowledge about ASD and typical development is said to be explored: what are the topics asked about? it seems to be the most interesting part and it is hidden.

The phrase "a total of 769 participants..." should belong in the "methods" section. The sentence about the salary category is not clear whether it is "high" or "low". In general, in the abstract there is no idea linking the knowledge and attitudes of the general population about ASD with the impact on early diagnosis (then in the introduction it is justified but it should be clear in the abstract, since it is the justification of the work).

In my opinion, there is a lack of detail on the questions in the questionnaire, both on knowledge and attitudes. The questions on attitudes appear in the graphs of the results but not those about knowledge of ASD, so the reader cannot know which aspects the population is being asked about, a crucial part of the study.

In the results there seems to be an excess of tables. For example, Table 1 could be replaced by text presenting the main percentages. Table 2 does not reflect the aspects of the disorder that have been shown to be better known among the general population (related to the fact that they are not detailed in the method either).

Sections 3.3. and 3.4. of the results seem to be presented in a more understandable format. I suggest unifying both graphs (why don't the graphs follow the same format? review apa format) and consider making a similar graph for section 3.2. on knowledge where the questions of the questionnaire and the response can be appreciated.

The discussion is difficult to assess when the reader does not know what aspects have been asked about the disorder, and therefore, to what extent it is relevant whether the population knows about them or not.

In general terms, I believe that a questionnaire is intended to cover too many aspects, and each of the sections should be an instrument in itself with the detail, validation and justification necessary in a rigorous study. By presenting so much information, the objective of the work is diluted, and it seems that the aspects evaluated are not being treated with the rigor that they should be.

In the conclusions I do not see the relationship between the first sentence and the following suggestions. Although the aspects suggested about providing more funds for training etc., are of great importance, I do not see that it is a conclusion of the work in which the opinions and attitudes of the general population have been evaluated. I suggest providing only implications that are derived from the research conducted.

Line 16. Explored

Phrase line 27 and 28 is about traits of the disorder so it should go before in that paragraph.

Careful with categorical language: "refuse to change"-> I would remove "refuse".

line 41: It is, however,-> The diagnosis is, however (there is no certainty that the condition is four times greater, only statements about the diagnosis can be made).

line 45. there seems to be a missing period separating two sentences.

line 48: remove "simply".

Line 83: caregivers of the child (do not restrict to parents).

line 83-84, take care of the gender of the language "she verifies"? who?

… etc.

Please review the English wording as it contains numerous grammatical inaccuracies.

Author Response

First of all, we would like to thank the reviewer for the time and efforts in reviewing our manuscript. Kindly find below our response to your comments, which are tracked in the resubmitted draft.

The summary introduces from the beginning the acronym ASD without explaining its meaning. In the first sentence "diagnosis" is repeated.

- We have now addressed this comment, lines 11-12.

Knowledge about ASD and typical development is said to be explored: what are the topics asked about? it seems to be the most interesting part and it is hidden.

- Thank you for this valuable comment. We have now provided complete details concerning the topic discussed about ASD knowledge and attitude in the method section, lines 142-153.

The phrase "a total of 769 participants..." should belong in the "methods" section. The sentence about the salary category is not clear whether it is "high" or "low". In general, in the abstract there is no idea linking the knowledge and attitudes of the general population about ASD with the impact on early diagnosis (then in the introduction it is justified but it should be clear in the abstract, since it is the justification of the work).

- We have now addressed the above mentioned comments in the abstract, lines 11-17, 19-20, and line 26.

 In my opinion, there is a lack of detail on the questions in the questionnaire, both on knowledge and attitudes. The questions on attitudes appear in the graphs of the results but not those about knowledge of ASD, so the reader cannot know which aspects the population is being asked about, a crucial part of the study.

- Thank you for this valuable comment. We have now provided complete details concerning the topic discussed about ASD knowledge and attitude in the method section, lines 142-153.

In the results there seems to be an excess of tables. For example, Table 1 could be replaced by text presenting the main percentages. Table 2 does not reflect the aspects of the disorder that have been shown to be better known among the general population (related to the fact that they are not detailed in the method either).

We preferred to offer the entire details of the study participants' demographic characteristics in Table 1 and only write the main findings in the main text, as this is the most common manner of presenting the study participants' demographic data in the majority of publications. Table 2 shows the mean knowledge score of the participants stratified by demographic factors, as mentioned in the method section (second section of the questionnaire, lines 140-143). We didn't want to present the answers to each knowledge item separately because that wouldn't give us a complete picture of our study participants' knowledge levels. However, we included the overall mean knowledge score because it is more informative for their current level of ASD understanding.

Sections 3.3. and 3.4. of the results seem to be presented in a more understandable format. I suggest unifying both graphs (why don't the graphs follow the same format? review apa format) and consider making a similar graph for section 3.2. on knowledge where the questions of the questionnaire and the response can be appreciated.

- We would like to thank the reviewer for this valuable comment. We have now used the same graph format based on the reviewer comment. Concerning section 3.2. on knowledge, the format of these questions was not the same (was not 5-point Likert scale) but was yes/no format. Based on the reviewer comment. We have now added a new graph (Figure 1) show the percentage of participants who answered each questions by yes/no along with the knowledge scale questions.

The discussion is difficult to assess when the reader does not know what aspects have been asked about the disorder, and therefore, to what extent it is relevant whether the population knows about them or not.

 - Based on the reviewer comment, we have now added in the method section complete details concerning the topic discussed about ASD knowledge and attitude, lines 142-153. Which are now discussed in the discussion section in pages 10-12.

In general terms, I believe that a questionnaire is intended to cover too many aspects, and each of the sections should be an instrument in itself with the detail, validation and justification necessary in a rigorous study. By presenting so much information, the objective of the work is diluted, and it seems that the aspects evaluated are not being treated with the rigor that they should be.

 - We would like to thank the reviewer for this valuable comment. As we have mentioned in the method section, we have used previously used and validated questionnaire tool in order to increase the reliability of our used tool and to compare our findings to previous populations. Despite that our questionnaire tool is exploring more than one dimension such as knowledge and attitude, the whole questionnaire tool is not very long to be completed or presented in the manuscript. We presented the knowledge as a continuous variable (mean knowledge score) and graphically (as per your recommendation) and discussed our findings in the discussion section. Similarly, we presented our findings concerning the attitude graphically and discussed it. However, we have now improved the writing in the discussion section to make it more clear.

In the conclusions I do not see the relationship between the first sentence and the following suggestions. Although the aspects suggested about providing more funds for training etc., are of great importance, I do not see that it is a conclusion of the work in which the opinions and attitudes of the general population have been evaluated. I suggest providing only implications that are derived from the research conducted.

 - Thank you for this valuable comment. We have now re-wrote the conclusion and focused on the aspects highlighted by the reviewer and provided implications based on area of deficiency in public knowledge about ASD which are knowledge about ASD treatment and etiology.

Line 16. Explored

- We have now addressed this comment, line 21.

Phrase line 27 and 28 is about traits of the disorder so it should go before in that paragraph.

- We have now addressed this comment and rephrased the whole paragraph, lines 27-34.

Careful with categorical language: "refuse to change"-> I would remove "refuse".

- We have now rephrased the sentence to address the reviewer comment, lines 49-50.

line 41: It is, however,-> The diagnosis is, however (there is no certainty that the condition is four times greater, only statements about the diagnosis can be made).

- We have removed the sentence to prevent any confusion for the reader.

line 45. there seems to be a missing period separating two sentences.

- We have now addressed this comment and rephrased the sentence, lines 54-57.

line 48: remove "simply".

- We have now addressed this comment, line 62.

Line 83: caregivers of the child (do not restrict to parents).

- We have now addressed this comment, line 97.

line 83-84, take care of the gender of the language "she verifies"? who?

… etc.

 - We have now addressed this comment and rephrased the sentence to make it clear, lines 96-99.

Please review the English wording as it contains numerous grammatical inaccuracies.

- We have now addressed this comment.

Reviewer 2 Report

Introduction:

  1. In lines 65-67, the ADOS (Autism Diagnostic Observation schedule) should also be mentioned as the gold standard tool for making an autism diagnosis.
  2. Paragraph 69-75 is written with poor grammar and should mention additional risk factors (age of father and neurotoxicant exposure)

Methods and Results

  1. The fact that 86.1 percent of the sample was male should be reported as a limitation of the investigation.
  2. The authors report that "Overall, the study participants showed weak level of knowledge about autism with 147 a mean score of 5.9 (SD: 3.1), comprising 34.7% of the total maximum obtainable score". It would be most advantageous to report data on "autism knowledge" from other cultures and countries in the introduction. Is the global knowledge of autism also this low or does it vary significantly by country?

Discussion:

1. In lines 211-217 there is some mention of similar  survey studies that were conducted in Australia and Africa. I believe it is essential to the value of this paper for a more extensive introduction and discussion on this top, presenting more studies and precise findings along with statistics. As a reader, I am left feeling that the presentation of the current results is incomplete without this broader discussion taking place.

Author Response

First of all, we would like to thank the reviewer for the time and efforts in reviewing our manuscript. Kindly find below our response to your comments, which are tracked in the resubmitted draft.

Introduction:

  1. In lines 65-67, the ADOS (Autism Diagnostic Observation schedule) should also be mentioned as the gold standard tool for making an autism diagnosis.

- We would like to thank the reviewer for this valuable comment. We have now addressed the above mentioned point in lines 87-90.

  1. Paragraph 69-75 is written with poor grammar and should mention additional risk factors (age of father and neurotoxicant exposure)

- We have now addressed the reviewer comments and re-wrote the paragraph to make it easier for the reader, lines 96-108.

Methods and Results

  1. The fact that 86.1 percent of the sample was male should be reported as a limitation of the investigation.

- Thank you for this valuable comment. We have now added this point to the limitations of the study, lines 380-381.

  1. The authors report that "Overall, the study participants showed weak level of knowledge about autism with 147 a mean score of 5.9 (SD: 3.1), comprising 34.7% of the total maximum obtainable score". It would be most advantageous to report data on "autism knowledge" from other cultures and countries in the introduction. Is the global knowledge of autism also this low or does it vary significantly by country?

- We would like to thank the reviewer for this valuable comment. We have now discussed this point in the discussion section, lines 268-275, 277-283 and 297-301 and highlighted that many variables contribute to the difference in the level of public knowledge about ASD between countries, these include cultural differences and information sources (whether internet, social media or TV).

Discussion:

  1. In lines 211-217 there is some mention of similar survey studies that were conducted in Australia and Africa. I believe it is essential to the value of this paper for a more extensive introduction and discussion on this top, presenting more studies and precise findings along with statistics. As a reader, I am left feeling that the presentation of the current results is incomplete without this broader discussion taking place.

- Thank you for this valuable comment. We have now discussed our findings further and added more studies. However, as there are different tools that examined the knowledge about ASD between different studies we presented discussed our findings in more general way without mentioning the mean score (as it will be not comparable to other studies), lines 268-275, 277-283 and 297-301.